# The effect of echoes interference on phonon attenuation in a nanophononic membrane

Mohammad Hadi [1,5], Haoming Luo [1,2,3,5], Stéphane Pailhès[1], Anne Tanguy [2], Anthony Gravouil[2], Flavio Capotondi [4], Dario De Angelis [4], Danny Fainozzi [4], Laura Foglia [4], Riccardo Mincigrucci [4], Ettore Paltanin [4], Emanuele Pedersoli [4], Jacopo S. Pelli-Cresi[4], Filippo Bencivenga[4] & Valentina M. Giordano [1] ✉

Nanophononic materials are characterized by a periodic nanostructuration, which may lead to coherent scattering of phonons, enabling interference and resulting in modified phonon dispersions. We have used the extreme ultra-violet transient grating technique to measure phonon frequencies and life-times in a low-roughness nanoporous phononic membrane of SiN at wavelengths between 50 and 100 nm, comparable to the nanostructure lengthscale. Surprisingly, phonon frequencies are only slightly modified upon nanostructuration, while phonon lifetime is strongly reduced. Finite element calculations indicate that this is due to coherent phonon interference, which becomes dominant for wavelengths between ~ half and twice the inter-pores distance. Despite this, vibrational energy transport is ensured through an energy flow among the coherent modes created by reflections. This inter-ference of phonon echos from periodic interfaces is likely another aspect of the mutual coherence effects recently highlighted in amorphous and complex crystalline materials and, in this context, could be used to tailor transport properties of nanostructured materials.

Controlling heat (transport, storage and conversion) is a major societal challenge, which can reduce the ecological footprint of our lifestyle, by recovering energy from heat losses and by increasing the device efficiency[1]. Heat propagates through lattice waves, whose corpuscolar equivalent is called phonon. The control of heat flow through nanostructuration has, until now, mostly exploited the phonon particle nature through the introduction of interfaces which incoherently scatter phonons with wavelength comparable to the nanostructure lengthscale. Exploiting its wave nature is still at its infancy and is the goal of nanophononics[2]: the design of materials periodically nanostructured allows to tailor thermal transport at the scale of phonon wave properties, whose coherence is preserved, opening new exciting perspectives in thermal management and

energy harvesting[3–7], but also in information transfer and quantum computing[8,9]. Indeed, in presence of a periodicity, phonons can be coherently scattered, keeping their phase and interfering, leading to the definition of a new Brillouin zone and modified phonon dispersions[3,10–16], directly impacting their contribution to thermal transport. And indeed, an important thermal conductivity reduction has been reported[3,14,15]. However, one aspect deserves further inves-tigations: the role of coherent mechanisms, such as phonon inter-ference, in determining the phonon lifetime ($\tau$) in such structured materials and how this relates to energy trasport. Recently, the relevance of coherent interactions between densely packed phonon branches has been evidenced in amorphous and complex crystalline materials: energy would hop among short-living modes thanks to

[1]University of Lyon, Université Claude Bernard Lyon 1, CNRS, Institut Lumière Matière, F-69622 Villeurbanne cedex, France. [2]LaMCos, INSA-Lyon, CNRS UMR5259, Université de Lyon, F-69621 Villeurbanne Cedex, France. [3]LMS, CNRS, École Polytechnique, Institut Polytechnique de Paris, 91128 Palaiseau, France. [4]Elettra Sincrotrone Trieste S.c.P.A., Strada Statale 14, km 163.5, AREA SCIENCE PARK, I-34149 Basovizza, Trieste, Italy. [5]These authors contributed equally: Mohammad Hadi, Haoming Luo. ✉e-mail: valentina.giordano@univ-lyon1.fr

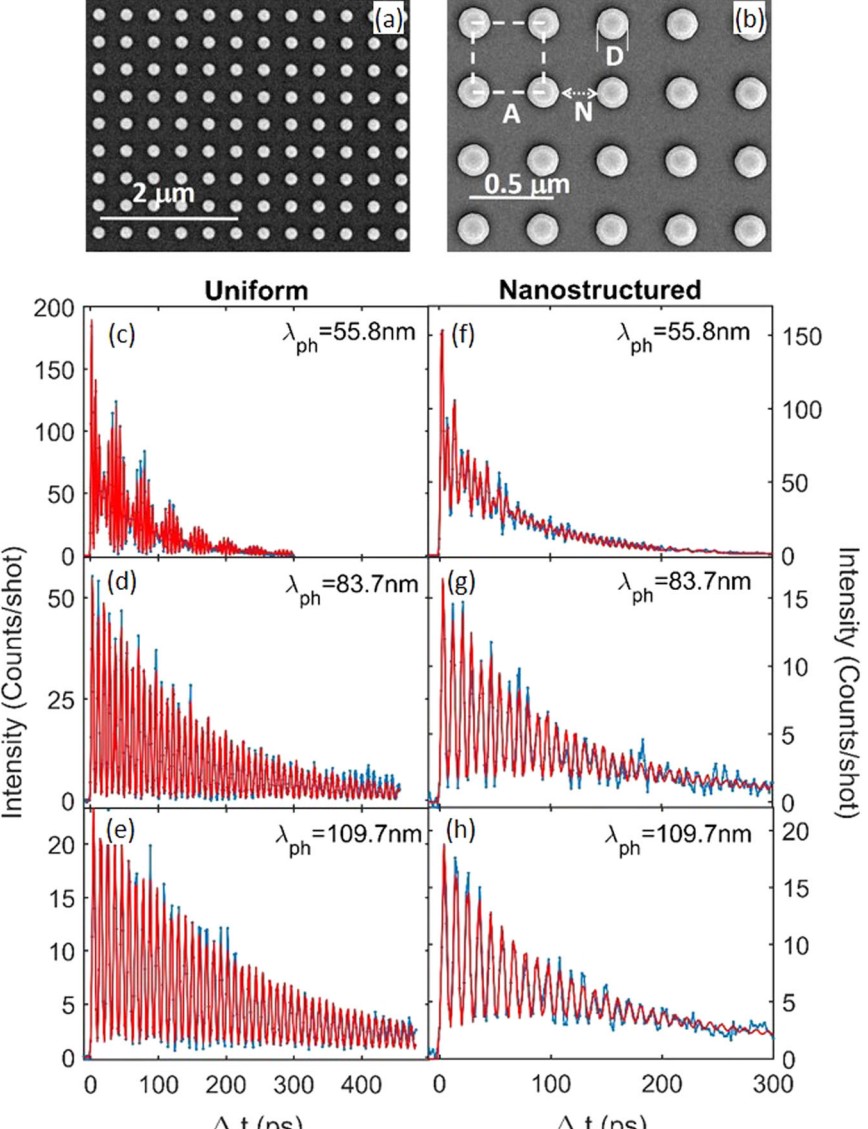

**Fig. 1 | Nanostructured sample and EUV TG signal from a uniform (U) and a nanostructured (NS) SiN membrane. a, b** TEM image of the nanoporous SiN membrane. The spatial scale of 2 $\mu$m (**a**) and 0.5 $\mu$m (**b**) are indicated by the white segments. The conical shape (see Methods) of the holes can be perceived in the latter image, with the inner diameter of the hole appearing as a white circle. The square of dashed lines represents the unit cell of the pores lattice, with indicated the period $A = 377$ nm, the neck $N = 253$ nm and the diameter $D = 124$ nm. **c–h** The experimental waveforms (blue dots connected with lines) are reported as a function of $\Delta t$; the values of $\lambda_{ph} = L_{TG}$ are reported in the individual panels. Red lines are best fits, which account for both thermal relaxation and a variable number of phonon modes (see the text). Notice that the range in $\Delta t$ for the NS is much shorter than for the U.

their phase correlation[17–20]. In a nanophononic material, we could expect a similar phenomenology among reflected waves. Still, despite simulations have clearly shown the change of phonon character from propagative to diffusive to localized depending on its wavelength, the connection with interference has not been further investigated[21–23]. On the experimental side, research drags behind, mainly due to the technical difficulty of accessing phonons with wavelengths ($\lambda_{ph}$) comparable to the nanostructure lengthscale, which, in many cases, are as short as a few tens of nanometers. Such range lies beyond the capabilities of standard phonon spectroscopies, such as Brillouin light scattering, limited to values of $\lambda_{ph}$ larger than a few hundreds of nanometers, or inelastic scattering of X-rays and neutrons, which, conversely, are used for $\lambda_{ph} < 10$ nm. Pump-probe approaches like picosecond ultrasonics can in principle access this wavelength range, but are typically limited to phonons propagating orthogonal to the sample surface[16], while pump-probe in an all-optical transient grating configuration can probe in-plane

phonons, but are typically limited to $\lambda_{ph} > 1\mu m$[24,25]. As a consequence, thermal transport theories have built mainly upon thermal conductivity measurements[26–32] dramatically needing experimental evidence on the coherent effect of the periodic nanostructure on phonons with wavelengths comparable to its lengthscale.

In this work, we present the experimental measurement of phonon lifetime at wavelengths across the 100 nm range in a nanoporous phononic membrane, by means of the extreme ultraviolet transient grating technique (EUV TG). In order to unveil the effect of coherent interference on phonon lifetime, we have studied phonons with wavelength comparable to the nanostructure lengthscale (pitch $A$ and neck $N$ in Fig. 1), and mean free path $\ell > (A, N)$, so that interface scattering is expected to dominate. For this, we have chosen to work with suspended membranes of amorphous SiN, a technologically relevant material[33–38], in which long lifetimes leading to mean free paths larger than $1\mu m$ have been predicted for phonons in the sub-THz range, i.e. the one here investigated[39].

By comparing the results from the nanoporous membrane with those obtained on a non-nanostructured membrane, we demonstrate that, in the nanophononic sample $\tau$ is reduced by a factor from 3 to 10, and exhibits a different dependence on phonon frequency ($\nu$). Combining our experimental results with finite element simulations, we evidence the presence of a dominant coherent mechanism, which accounts for up to 65% of phonon attenuation in the nanostructure at room temperature and for $\lambda_{ph} \sim 100$ nm. While coherent interference inhibits the individual phonon propagation, the energy flow keeps going on, thanks to an energy transmission among the coherent modes created by reflections and interference. As such, we show the evidence of an interference coherence effect analog to the mutual coherence effect highlighted in complex crystals and amorphous materials[17–20].

## Results and discussion

This study was performed on a uniform (U) and a nanostructured (NS) suspended membrane of LPCVD amorphous SiN, with thickness $d = 55$ nm. The nanostructure consists in a square lattice of pores with $A = 377$ nm, average diameter $D = 149 \pm 7$ nm and neck $N=253$ nm (Fig. 1). The nominal density and Young modulus were $\rho_0=2.9$ g/cm$^3$ and $E = 270 \pm 25$ GPa, respectively, as provided by the supplier (see Methods).

In order to investigate the dynamics of phonons with $\lambda_{ph}$ comparable to such a nanostructure, we have used the EUV TG technique[40], where the interference between two EUV pulses within the sample generates a spatially modulated excitation with periodicity $L_{TG}$, equal to the wavelength of the generated coherent phonons. In our case it was $L_{TG} = 55.8$, 83.7 and 109.7 nm. The picosecond dynamics of the excited sample was determined by transient diffraction of a third EUV pulse (probe), with a variable time delay $\Delta t$. More details are in Methods and in the Supplementary Section I.

The EUV TG signal collected at the three values of $L_{TG}$ from both membranes is reported in Fig. 1. The observed waveforms are the result of a thermal relaxation at $k = 2\pi/L_{TG}$ due to the impulsive heating of the sample and coherent phonon dynamics[25,40] and may be modeled as[40]:

$$I_{TG}(\Delta t) = |A_{th}e^{-\Delta t/\tau_{th}} + \Sigma_i A_i e^{-\Delta t/\tau_i} \cos(2\pi\nu_i\Delta t)|^2 \qquad (1)$$

Here the first term accounts for the thermal relaxation, with amplitude $A_{th}$ and characteristic time $\tau_{th}$. The second term describes the sum of coherent oscillations (phonons) with frequency $\nu_i$, whose amplitude decreases exponentially with time from its initial value $A_i$ due to the finite phonon lifetime $\tau_i$. Looking at the figure, the most evident result is the large reduction in phonon lifetimes for the NS sample: the coherent oscillations last much shorter than in the uniform membrane.

In the following, we focus on phonon properties, while thermal relaxation is discussed in the Supplementary Section III.

### Phonon dispersions

In a confined material, like a membrane, boundary conditions result in so-called waveguided phonons, propagating in plane along the membrane and symmetric or antisymmetric with respect to the center of the membrane thickness. Two kinds of waves exist: shear waves, with an in-plane polarization (Love waves), and waves with a mixed longitudinal and shear off-plane polarization (Lamb waves). In our EUV TG experiment, symmetric Lamb modes with $\lambda_{ph} = L_{TG}$ and predominant longitudinal polarization are expected to have the largest cross-section[25,41].

We can obtain a good fit of the waveforms reported in Fig. 1 using for both samples 2 phonons at $L_{TG}=83.7$ nm and 1 phonon at $L_{TG}=109.7$

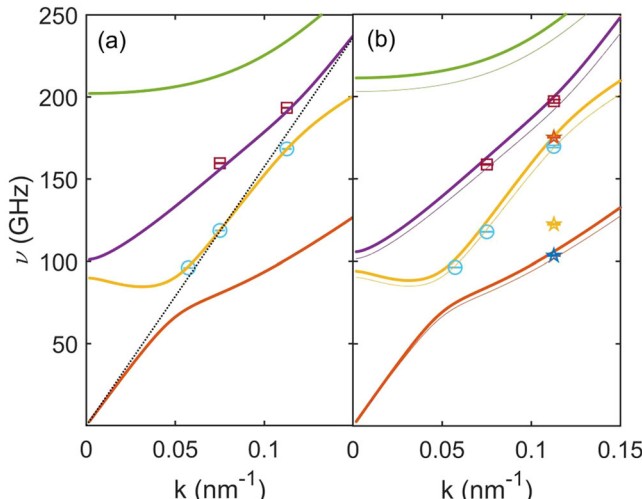

**Fig. 2 | Experimental and calculated phonon frequencies.** The values of $\nu$, as extracted from the best fit of experimental data through Eq. (1), are reported as a function of $k = 2\pi/\lambda_{ph}$ for the uniform membrane in (**a**) and the nanostructured one in (**b**), together with the dispersions for symmetric Lamb waves calculated for a uniform membrane, using a Young modulus of 227.5 GPa, a Poisson ratio of 0.27 and a reduced average density $\rho_{NP} = 2.653$ g/cm$^3$ for the nanostructured material (thick solid lines in (**a**) and (**b**)). Thin lines in (**b**) are calculated by using a Young modulus of 210 GPa. Symbols identify the phonon branch for experimental data: blue circles for the $S^2$ branch, red squares for $S^3$, while stars are the phonon modes of the nanostructure that have no counterparts in the uniform sample. Error bars come from the fit uncertainties. The black dotted line in (**a**) is the longitudinal phonon acoustic dispersion expected in a bulk sample.

nm, while a different number of phonons for the two samples is needed at $L_{TG} = 55.8$ nm: 2 in the uniform membrane and up to 5 in the nanostructured one (see Supplementary Section II for the fit procedure and all fitting parameters). In the nanophononic membrane, at all probed values of $\lambda_{ph}$, minor features in the Fourier transform suggest the presence of additional modes with a smaller amplitude. However, we have chosen to limit the number of fitting parameters and consider only the main Fourier components.

The values of $\nu_i$, as obtained from the fit, are reported in Fig. 2a and b for the uniform and the nanostructured membrane, respectively. The experimental frequencies for the uniform sample are compared with the analytic calculation of symmetric Lamb modes: we find a good agreement using $\rho_0$ and the nominal Poisson ratio $\nu = 0.27$ but a Young Modulus smaller than the nominal one, $E = 227.5$ GPa, resulting in longitudinal and transverse velocities $v_L = 9.9(1)$ Km·s$^{-1}$ and $v_T = 5.56(1)$ Km·s$^{-1}$. The measured frequencies precisely fall onto the second ($S^2$) and the third ($S^3$) symmetric mode branches, which are those closer to the longitudinal phonon frequency of a bulk sample.

In Fig. 2b, we compare the experimental values of $\nu_i$ from the nanostructured sample with the ones calculated for a uniform membrane but with a reduced average density ($\rho_{NP} = 0.915\rho_0 = 2.653$ g/cm$^3$), due to the porosity. We can recognize the same phonon modes detected in the uniform membrane, with slightly lower frequencies. This phonon softening is an expected effect of porosity, which is known to reduce the Young modulus[42]. Indeed, the agreement improves by using $E = 210$ GPa, while we notice three frequencies, marked with stars in Fig. 2b, that escape this model. Overall the agreement for the NS sample is remarkably good for a model which is clearly too simplistic, as it assumes a uniform and isotropic medium and cannot account for the presence of the square pore lattice, thus disregarding phonon dispersion modifications. The three additional modes observed in the NS sample at the largest value of $k$ (corresponding to $\lambda_{ph} = 55.8$ nm) are likely related to such effects.

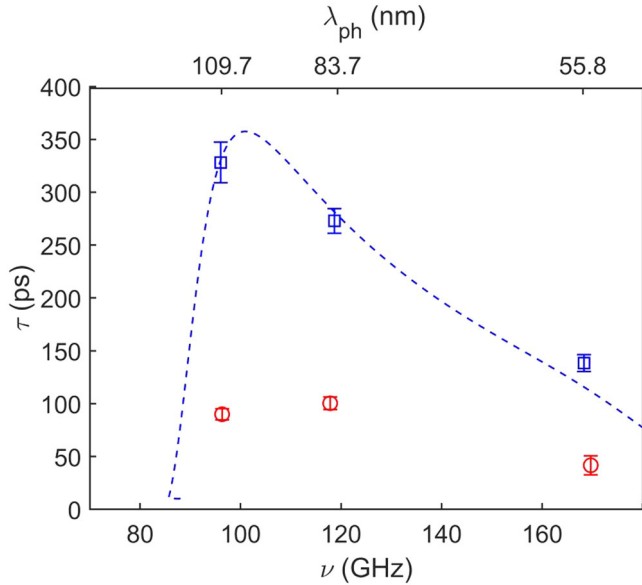

**Fig. 3 | Experimental phonon lifetimes for branch $S^2$.** Blue squares and red circles represent, respectively, data belonging to the uniform and NS membranes. The blue dashed line is the isothermal lifetime calculated using the result of the fit of the data for the uniform membrane taking into account both anharmonic and surface scattering (see the text). Error bars on experimental data come from the fit uncertainties. The corresponding wavelengths are reported in the upper horizontal axis.

## Phonon decay

Once we have identified the phonons belonging to the same Lamb modes in the two samples, we compare their lifetime $\tau$ as a function of frequency $\nu$ in Fig. 3 for the $S^2$ branch. We observe, for a given value of $\nu$, a sizable reduction in the value of $\tau$ when going from the uniform to the nanostructured membrane, by a factor ≈ 3. An even larger reduction, by almost an order of magnitude, is observed for the $S^3$ branch, for which however we have only 2 points (see Supplementary Fig. 2).

In amorphous SiN, in the probed wavelength range, anharmonic phonon-phonon scattering is expected to be the dominant process for phonon decay[39]. However, we also expect an important contribution from boundary scattering from the membrane surfaces as $\lambda_{ph}$ is comparable with $d$. We model the former using the Akhiezer approach, which has been found to well describe phonon decay in SiN at THz frequencies[39] and gives $\tau_{ph-ph} = \frac{v^3}{BT\nu^2}$. Here $v$ is the phonon group velocity and $B$ a constant related to various material properties. We model the contribution to phonon decay due to incoherent boundary scattering following Ziman[43]: $\tau_b = \frac{1+p}{1-p}\frac{d}{v}$, with $\eta$ the roughness and $p = e^{-16\pi^2\eta^2/\lambda_{ph}^2}$ the specularity parameter. The total phonon lifetime is finally calculated using the Mathiessen rule: $\frac{1}{\tau_U} = \frac{1}{\tau_{ph-ph}} + \frac{1}{\tau_b}$. We have fitted the data from the uniform membrane using this model, with $B$ and $\eta$ as free parameters. The best fit (dashed blue line in Fig. 3) is obtained with $B = 1.7 \pm 0.6 \times 10^{-4}\,\text{m}^3\text{s}^{-2}\text{K}^{-1}$ and $\eta = 0.6 \pm 0.1$ nm, the latter value is larger than the one measured by AFM on the top surface of our membranes ($\eta = 0.41 \pm 0.05$ nm), likely due to the sensitivity of the model to the roughness of both membrane surfaces. As reported in the Supplementary Section V, these two mechanisms lead to similar contributions to $\tau_U$.

In the NS sample one can expect the same mechanisms plus the effect of the nanostructure, which may lead to coherent (specular reflections and interference effects) and incoherent (scattering from hole boundary roughness) contributions, as well as to a decrease in $\tau_{ph-ph}$ due to phonon branch folding, which leads to the appearance of lower energy, less dispersive optical modes, favorable for anharmonic

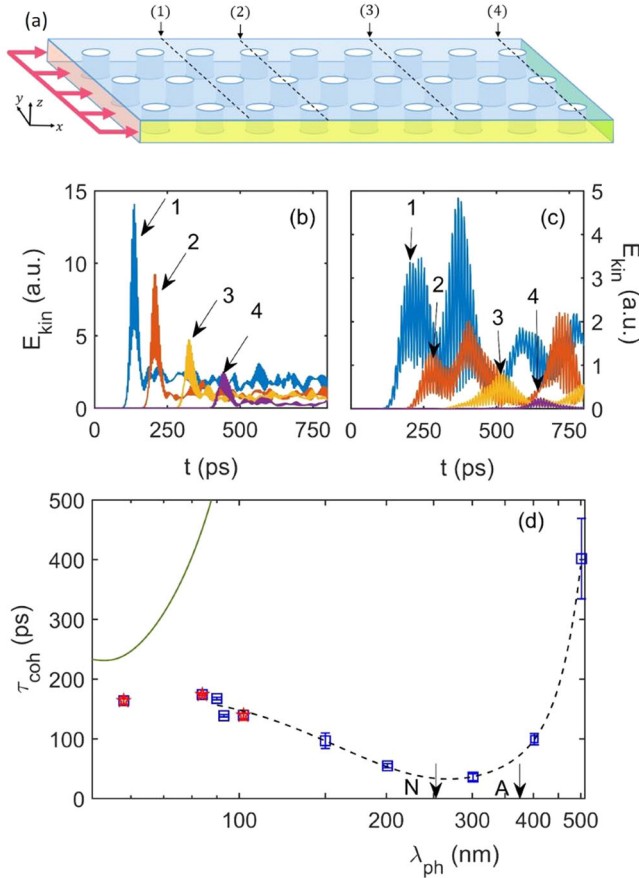

**Fig. 4 | Wavepacket propagation and phonon lifetime as simulated by finite element modeling. a** Sketch of the 3D finite element model of the nanophononic structure. A semi-infinite solid is constructed by applying Periodic Boundary Conditions (yellow sides) and Perfect Matched Layers (end of the sample on the right). The wavepacket is generated on the left side of the sample (red). The four $x$ positions at which the time evolution of the wavepacket is reported in (**b**) and (**c**) are indicated with dashed lines. **b**, **c** report the time dependence of the instantaneous kinetic energy $E_{kin}(x, t)$ at the 4 selected positions, identified by the different colors (blue, orange, yellow, magenta from 1 to 4), for $\lambda = 84$ nm and $\lambda = 201$ nm, respectively. Arrows indicate the first peak in time that reaches each of the four positions. The coherent phonon attenuation was calculated from the decay of this peak and is reported in (**d**) as a function of the wavelength $\lambda_{ph}$ in a logarithmic scale. Simulated points at the experimental wavelengths are reported in red. The black dashed line is a guide for the eye. The green solid line is the incoherent boundary scattering lifetime $\tau_b$. The neck (N) and periodicity (A) are indicated by arrows. Error bars come from the fitting uncertainties.

phonon-phonon scattering[15,44]. We included the incoherent scattering from hole boundary and the increased anharmonic scattering in $\tau_b$ and $\tau_{ph-ph}$, respectively, by assuming an effective roughness ($\eta_{NS}$)[28] and $B_{NS} > B$; it's worth mentioning that we verified by AFM that the surface roughness in the NS sample is the same as in the uniform one, meaning that an increase in effective roughness can thus be ascribed to hole boundary. We thus model the phonon lifetime in the nanostructure as:

$$1/\tau_{NS} = 1/\tau_{ph-ph} + 1/\tau_b + 1/\tau_{coh} \qquad (2)$$

with $\tau_{coh}$ the coherent contribution due to the nanostructure.

In order to estimate this latter, we have run finite element calculations of the propagation of acoustic wavepackets in a 3D membrane with a lattice of cylindrical pores having the same average values of $A$ and $D$ as in the present experiment (Fig. 4). Namely, we have created an impulsive quasi-monochromatic excitation with longitudinal polarization and followed its propagation through the sample along the

longitudinal position ($x$-axis) by monitoring the kinetic energy summed over the cross section $y - z$, $E_{kin}(x, t)$, as a function of $x$ and time $t$. Specifically, we define two quantities (see Supplementary Section IV): the instantaneous $E_{kin}(x_i, t)$ at selected positions $x_i$ along the propagation direction, and the maximum over all times of $E_{kin}$ as a function of $x$, $P_\nu(x) = \max_t E_k(x, t)$, that we call the envelope of the kinetic energy[21–23]. The former allows us to follow the time evolution of the wavepacket and see the residual energy at $x_i$ after its passage, while the latter allows us to follow the propagation of the total vibrational energy contained in the sample due to the wavepacket excitation. Details on the theoretical methods are given in Methods and in the Supplementary Section IV. As an example, we report in Fig. 4b and c $E_{kin}(x_i, t)$ for $\lambda_{ph} = 84$ nm and $\lambda_{ph} = 201$ nm at 4 different positions $x_i$, located from 1 to 3 $\mu$m distance from the location where the original wavepacket was generated. Here we can see the wavepacket propagating from position to position: the farther is $x_i$ from the origin, the later in time the first peak appears. Its decreasing amplitude indicates how it looses a relevant fraction of energy during the propagation. This energy loss is related to nanostructuration, since no other phonon decay mechanisms (anharmonicity and roughness) were implemented in the calculations, so that in a uniform membrane there would be no attenuation. We can understand it as due to a redistribution of the kinetic energy in directions different from the $x$-axis, so that an effective reduction of $E_{kin}(x, t)$ along $x$ appears. Interestingly, at later times there is still energy at a given $x$ position: it comes from the backreflections, which redistribute the vibrational energy in space, leading to a non structured residual kinetic energy at later times for $\lambda_{ph} = 84$ nm. The situation is different for $\lambda_{ph} \geq \sim 100$ nm (see Fig. 4c and Supplementary Figs. 6, 7 and 8): here the first peak in time, which corresponds to the fraction of the original wavepacket which continues to propagate along $x$, rapidly looses intensity, while interference between reflected wavepackets can produce later times peaks even

more intense than it. This is the result of a coherent interference between reflected wavepackets, which can interfere with each other at later times (like "echoes"), since they do not follow a straight propagation and therefore arrive later at a given position. As pores are circular, we have a broad distribution of reflected wavepackets, coming from different angles and with different path length differences, so that interference can be constructive or destructive, thus enhancing or depleting the amplitude of the interference pattern. Moreover, if the coherence length of the wavepackets is longer than the path difference between the original propagating wavepacket and the reflected ones, they can also interfere. This could be the case of wavepackets reflected in near forward directions, which can interfere with the later time tail of the remaining fraction of the original wavepacket before it has completely moved on, adding to its attenuation. A more extended discussion of time propagation for 54 nm $\leq \lambda_{ph} \leq 500$ nm at up to 18 positions within the sample is reported in the Supplementary Section IV.

From the attenuation of the first peak as a function of time, we have extracted the theoretical lifetime for the coherent contribution due to the nanostructure, $\tau_{coh}$. The trend of $\tau_{coh}$ as a function of $\lambda_{ph}$ is shown and compared to $\tau_b$ in Fig. 4d. First, we notice that $\tau_{coh} << \tau_b$, pointing to a major role of coherent mechanisms in determining the nanostructure contribution to phonon attenuation. Second, it may be seen that $\tau_{coh}$ suddenly decreases for $\lambda_{ph} \geq 84$ nm, exhibiting a shallow minimum around $\approx 200$–$300$ nm, i.e. for $\lambda_{ph} \approx N$. For $\lambda_{ph} > N$, $\tau_{coh}$ increases again, in agreement with the expectation that, for wavelengths much larger than the nanostructure length-scale, these effects should weaken and the uniform membrane behavior should be asymptotically recovered.

In a first approximation, $\tau_{NS}$ can be estimated using Eq. (2) by considering the calculated values of $\tau_{coh}$ and the values of $\tau_{ph-ph}$ and $\tau_b$ observed in the uniform membrane. As can be seen in Fig. 5, the agreement with the experimental values is impressively good, clearly pointing to $\tau_{coh}$ as the main responsible for the observed lifetime difference between the two samples. As we also expect changes in $\tau_{ph-ph}$ and $\tau_b$, we fit our data with Eq. (2) with $B$ and $\eta_{NS}$ as free parameters. The best fit gives a slightly better agreement for $B = 2.7 \pm 0.8 \times 10^{-4} \mathrm{m}^3\mathrm{s}^{-2}\mathrm{K}^{-1}$ and $\eta_{NS} = 0.7 \pm 0.1$ nm, i.e. the expected increase in both anharmonicity and roughness. In order to quantify the importance of coherent mechanisms, we have calculated their relative contribution to the total phonon decay rate, finding that this increases from 36(2)% for $\lambda_{ph} = 55.8$ nm to 65(1)% for $\lambda_{ph} = 109.7$ nm (see Supplementary Material).

Further insights on the role of coherent effects can be gained from $P_\nu(x)$. As far as the interference peaks remain small, as for $\lambda_{ph} = 84$ nm, the energy envelope essentially corresponds to the first peak in time of $E_{kin}(x, t)$ at a given position. This is no longer the case if peaks later in time are more intense than the first one due to constructive interference: at that point, $P_\nu(x)$ corresponds to the maximum of the most intense interference peak at a given position $x$. As an example, we report in Fig. 6a the envelope $P_\nu(x)$ in a semi-logarithmic scale for our experimental wavelengths, and for $\lambda_{ph} = 201$ and 502 nm. The global behavior is linear with $x$: this corresponds to a propagative energy (heat) transport, and the inverse of the slope gives the mean free path $\ell_{env}$[22]. For a comment on the detailed shape of $P_\nu(x)$, we address the reader to the Supplementary Section IV. Here we focus on the mean free path, that we report in Fig. 6b, together with the one calculated from the propagation of the first peak in $E_{kin}(x_i, t)$, using its velocity ($v = dx/dt$) and the estimated $\tau_{coh}$ ($\ell_{coh}$). We can see that, for 102 nm $< \lambda_{ph} < 502$ nm, i.e. $0.4N \leq \lambda_{ph-ph} \leq 2N$, $\ell_{env}$ remains systematically larger than $\ell_{coh}$. This is clearly the effect of the strong constructive interference: the envelope gives the effective mean free path of the total vibrational energy in the material, which is not the same of the excited wavepacket. The coherent interference strongly decreases the wavepacket lifetime and mean free path, while it keeps the total energy

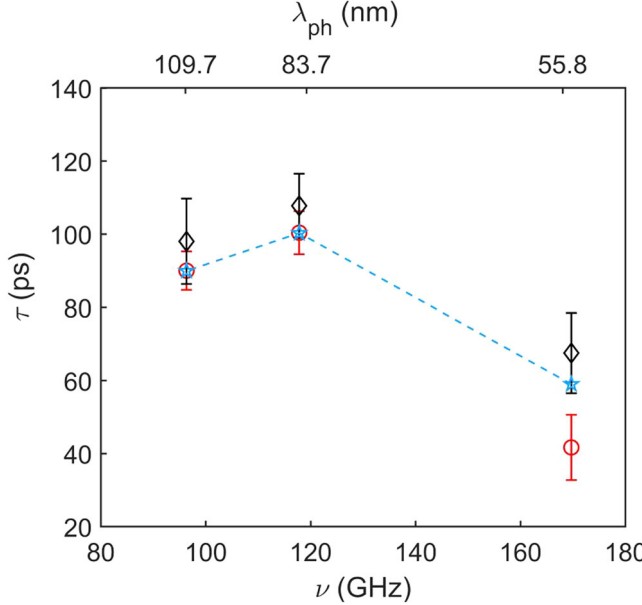

**Fig. 5 | Estimated and experimental phonon lifetime in the nanostructure for branch $S^2$.** Experimental data for the NS membrane (red circles) are compared with an estimation based on adding the simulated $\tau_{coh}$ to the lifetime of the uniform membrane (black diamonds); see Eq. (2). Light blue stars connected by dashed lines are the results obtained by adjusting $B$ and $\eta$ to account for different parameters in the NS sample (see text). Error bars on the experimental data come from the fit uncertainties, while for the estimation and the fit, they result from uncertainty propagation in Eq. (2). The corresponding wavelengths are reported in the upper horizontal axis.

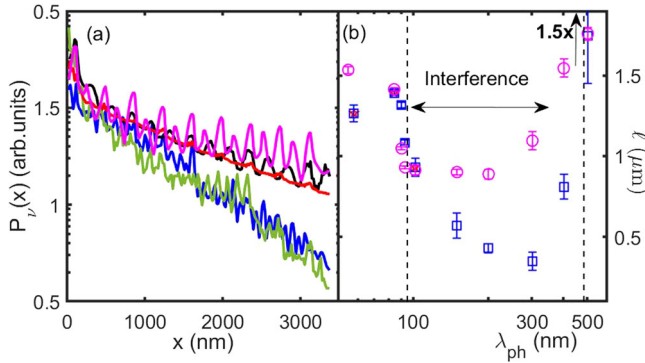

**Fig. 6 | Envelope of the kinetic energy simulated by finite element modeling.**
**a** We report in a semi-logarithmic scale the envelope of the kinetic energy for $\lambda_{ph}$=55 (black), 83 (red), 109 (blue), 201 (green) and 502 nm (magenta). In this scale, the linear dependence on the position $x$ corresponds to a propagative behavior with a mean free path given by the inverse of the slope. **b** The mean free path calculated for the envelope ($\ell_{env}$, magenta circles) is reported and compared with the one of the main pulse, calculated following its propagation ($\ell_{coh}$, blue squares), as a function of the wavelength in a logarithmic scale. Red stars correspond to this latter, for the experimental wavelengths. The two mean free paths depart from each other between 102 and 502 nm, due to an increased importance of coherent constructive interference between multiply scattered peaks in this wavelength range. Vertical dashed lines highlight this interference-dominated region. Note that the value for $\lambda_{ph}$ = 502 nm has been devided by 1.5 for a better visibility of all points. Error bars come from the fit of the mean free path for the envelope and of the lifetime for the main pulse.

much larger than the roughness cannot be thought of as a wavepacket with given energy and wavevector, but become a broad superposition of coherent modes, bouncing and propagating in many different directions, because of multiple reflections, and interfering with each other. Interference could be a powerfull knob to manipulate phonons at nanoscale wavelengths and can be used, e.g., for tailoring phonon-mediated thermal, electronic and magnetic properties of materials, as well as for developing new information technologies based on phonons to encode the logic function[8,9].

## Methods
### Sample
This study was performed on commercially available suspended membranes of LPCVD (low pressure chemical vapor deposition) amorphous SiN, provided by Norcada Inc. (Edmonton, Canada). The fabrication process of the nanostructured membrane has a specific directional selectivity in the material removal process, and it creates a sloped sidewall profile to the pores. The diameter resulted $149 \pm 7$ nm at the top surface and $99 \pm 7$ nm at the bottom surface, giving an average diameter $D$=124 nm. All of the device features were made with common semiconductor processing technology and tools.

As said in the main text, we have measured by AFM the roughness of the top surface of the uniform and nanostructured membranes and found a value $\eta = 0.41 \pm 0.05$ nm. In the fitting model, the roughness is taken into account in the $\tau_b$ term and fitted as a free parameter. The fitting result can be considered as an effective roughness, which empirically includes the contribution of top and bottom surfaces, as well as the walls of the pores, similarly to what proposed in ref. 28. The value of the effective roughness obtained by the fit (0.6-0.7 nm) is close to the measured one. It is reasonable to assume that the top surface is the flattest and, therefore, an effective roughness somewhat larger than the one measured in the top surface is not surprising even for the uniform membrane.

### Extreme utraviolet transient grating experiment
The extreme ultraviolet transient grating (EUV TG) technique was developed at the TIMER beamline of the FERMI free electron laser (Trieste, Italy)[40]. In this technique, two EUV pulses (pump) with the same wavelength $\lambda_{ex}$ in the [26-52] nm range are crossed onto the sample at an angle $2\theta = 27.6°$. Their interference generates a spatially modulated excitation in the sample with a periodicity $L_{TG} = \lambda_{ex}/2sin(\theta)$ ranging from 55.8 to 109.7 nm (see Supplementary Table I). The sample surface was orthogonal to the bisector of the crossed pump pulses and the surface of the NS sample has been oriented in order to have the [1 0] direction (in reciprocal space) of the pores square lattice parallel to the EUV TG wavector. The ps dynamics of the excited sample was determined by transient diffraction (detected in transmission) of a third EUV pulse (probe) at 13.3 nm wavelength, with a variable time delay $\Delta t$. For each step in $\Delta t$ we accumulated the EUV TG signal for about 1000-2000 laser shots (repetition rate 50 Hz); the tipical acquisition time for a single EUV TG waveform (shown in Fig. 1) was $\approx$ 3 hours.

In the employed experimental conditions, the pump penetration depth $L_{abs}$ is equal or shorter than the membrane thickness $d$, while the probe penetration depth ($\approx$ 120 nm) is larger. As such, probe diffraction is mainly caused by density modulations along $k$[45], so that symmetric Lamb modes with $\lambda_{ph} = L_{TG}$ and predominant longitudinal polarization are expected to have the largest cross-section[25,41]. It is worth noticing that the observation of symmetric Lamb modes, with no evidence of prominent antisymmetric modes, suggests negligible effects related to the finite penetration depth of the EUV pulses[45], while possible effects on the signal intensity are not relevant for the present study[25]. The finite penetration depth can instead affect cross-plane thermal transport, and, through possible cross talk between in-plane and cross-plane transport, it may affect the in-plane thermoelastic

transport almost unperturbed thanks to the energy transmission between reflected wavepackets and their constructive interference. Out of this interference region, the two mean free paths correspond, meaning that interference effects are less important, and phonon attenuation is mainly due to specular scattering at interfaces.

We have reported the first experimental determination of nanoscale phonon dynamics in a phononic nanoporous amorphous SiN membrane, for phonon modes with wavelengths comparable to the nanostructure lengthscale and we compared these results with those obtained from a uniform membrane. We find that, while phonon frequencies are only slightly softer in the nanoporous membrane, phonon lifetime is drastically reduced. By means of finite element calculations, we have shown that the observed behavior is the result of a coherent mechanism, which starts to become important already at $\lambda_{ph} \sim 100$ nm, i.e., phonon wavelengths of ~40% the neck, and dominates the wavepacket attenuation up to wavelengths twice the neck. Namely, the strong lifetime reduction is due to the redistribution of energy in time and space via multiple reflections and interference, with a stronger effect when the initial wavepacket interferes with its own reflections overlapping with it within its coherence length. Despite this, energy transport is almost unperturbed thanks to the mutual energy transmission among reflected waves, promoted by constructive interference. This coherent effect here reported can be likely regarded as another case of the mutual coherence recently proposed for amorphous and complex crystalline materials, where the coherence between different modes leads to a mostly diffusive heat transport[17,20]. Our findings go beyond the material actually investigated and will stimulate further developments in mutual coherence theories and advanced thermal transport models, able to account both for incoherent and coherent mechanisms in nanophononic materials. The identification of the wavelength region where this process dominates allows to target the nanostructure for interference-dominated phononic materials, opening new perspectives in a variety of application fields. Indeed, our data show how, in nanophononic materials, phonons with wavelengths comparable to the nanostructure and

dynamics here probed, in particular the thermal relaxation time $\tau_{th}$ of Eq. (1), as discussed in the Supplementary Section III.

The pump fluence $F$ was chosen in order to remain below the damage threshold and was a compromise between the necessity to have a bright signal and to limit the temperature increase ($\Delta T$) due to laser heating. We can estimate this latter from the pump energy absorbed by the sample within $L_{abs}$ and using the nominal density $\rho$ and literature data for the specific heat (C=700 J Kg$^{-1}$K$^{-1}$),[46]: $\Delta T = \frac{F(1-e^{-d/L_{abs}})}{\rho C d}$. Values of $\Delta T \sim 10 - 16$ °C have been found for both samples at $L_{TG}$=55.8 and 83.7 nm, leading to a final sample temperature $T_s = 25 + \Delta T \sim 35 - 40$ °C, while for $L_{TG}$=109.7 nm we estimated $T_s \sim 50 - 56$ °C. These estimates assume that the effective sample temperature is represented by the temperature rise in a volume limited by the pump's absorption length. As temperature enters in our modeling through the anharmonic scattering parameter $\tau_U$, this could thus affect our analysis. We have checked that, assuming different temperatures up to $\sim 100$ °C, fitting results are not significantly modified: it only leads to a slightly different value of the anharmonic parameter B ($B_U = 1.4 \pm 0.5 \times 10^{-4}$ m$^3$s$^{-2}$K$^{-1}$ and $B_{NS} = 2.23 \pm 0.07 \times 10^{-4}$ m$^3$s$^{-2}$K$^{-1}$ for the uniform and the nanostructured samples respectively). This does not change the physical interpretation of the role of coherent mechanisms on the phonon lifetime in the nanostructured membrane, since the dependence of $\tau_{NS}$ on $v$ cannot be explained if the $\tau_{coh}$ term is ignored. The main experimental parameters are summarized in Supplementary Table I.

## Theoretical modeling

In this study, we investigated the behavior of wave packets propagating through a 3D semi-infinite elastic nanostructured membrane, with a thickness of 55 nm as shown in Fig. 4a. The membrane consists of nine squares, of side 377 nm (periodicity of the system) and containing a single cylindrical pore with a diameter of 124 nm. The structure is duplicated along the y direction to form three rows of pores. Such a large domain allows us to observe propagation over a long time period. The material properties used in this study, including density (2.9 g cm$^{-3}$), Young's modulus (227.5 GPa) and Poisson ratio (0.27), are consistent with the ones of the experimenal study.

To introduce quasi-monochromatic propagating wavepackets, we apply a displacement on the left side of the model around $x = 0$ for a small time interval starting at $t \geq 0$:

$$U(\nu,t) = U_0 \exp\left[-\frac{(t-3t_0)^2}{2t_0^2}\right] \sin(2\pi\nu t), \qquad (3)$$

where $U_0$ is a constant, $v$ is the frequency of the quasi-monochromatic excitation, and $t_0 = \frac{3\pi}{2\pi\nu}$ is the coherence time of the wavepacket. Written in this way, the wavepacket starts to be generated at $t = 0$ and has a total time extension of about $6t_0$. In this study, we only consider a displacement perpendicular to the boundary, which corresponds to a longitudinal excitation. Top and bottom surfaces are free boundaries, while on the other directions we apply two types of boundary conditions, as shown in Fig. 4: periodic boundary conditions along y, and a perfect matched layer on the right side of the model, along x. This latter is used to absorb the incident wave and prevent reflection, ensuring that there isn't a spurious interference with the backreflected wave from the boundary.

Simulations have been run using the open source software Cast3m[47]. The 3D mesh is generated within Cast3m by extruding a 2D mesh along the Z-direction using linear elements, including brick and prism elements, with the prism elements primarily located around the holes. The mesh size has been carefully selected to ensure calculation precision while also maintaining affordable simulations lengths. For this study, a mesh with approximately $1.6 \times 10^6$ nodes ($4.8 \times 10^6$ degrees of freedom) is a satisfactory choice, as we explain below. To

accurately represent a wave on a discrete grid, it is generally required to have at least $12^3 = 1728$ degrees of freedom per volume measured in a wavelength cube in 3D, according to empirical criteria. For our case, the chosen mesh gives a density of degrees of freedom (e.g., total degrees divided by the volume) of 0.025 degrees of freedom per nm$^3$. In our case, the shortest wavelength is 58 nm, resulting in a total of $58^3 \times 0.025 = 48778$ degrees of freedom, which is much higher than the requirement of accuracy.

To obtain the temporal evolution of the propagating wavepacket, we used the symplectic central-difference time integration algorithm. The time step $\Delta t$ was chosen to satisfy the convergence condition: $\Delta t < \Delta t_{cr}$, where the critical time step $\Delta t_{cr} = \frac{\Delta l}{c_{wave}}$. Here, $\Delta l$ is linked to the mesh size, and $c_{wave}$ is the longitudinal wave speed. We used the same time step of simulation, $\Delta t \sim 4$ fs, for all frequencies considered in our study, and carried out simulations with a total of 20,000 time steps, except for the longest wavelength, 500 nm, for which we ran a total of 40,000 time steps. Technical details about the boundary conditions and the time integration scheme can be found in the Appendix of[22], with an example provided for 2D simulations. We have extended this approach to 3D in this work.

It is worth noticing that, despite the transient grating excitation extends over a region larger than our model, the only relevant direction is the one parallel to the transient grating wavevector. Therefore the dynamics along this direction (x-axis) and at a given value of $\lambda_{ph}$ are representative of such a mechanic perturbation and are monitored by the kinetic energy summed over the cross section $y - z$, $E_{kin}(x,t)$, as a function of $x$ and time $t$.

## Data availability

Processed experimental data generated in this study have been deposited in the Zenodo database under accession code DOI 10.5281/zenodo.10513797. Raw data are available in the open access Elettra data repository under accession code https://doi.org/10.34965/i11160. Simulation data are available upon request to the authors.

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

## Acknowledgements

V.M.G. and S.P. acknowledge support from IdexLyon (ANR-16-IDEX-0005) under the Scientific Breakthrough Project program for project IPPON (VMG,SP). V.M.G. acknowledges support from the ANR (project MAPS-ANR-20-CE05-0046,VMG). This project has received funding from the European Union's Horizon 2020 research and innovation program under grant agreement no. 654148 and 871124 Laserlab-Europe (VMG). H.L. and V.M.G. thank Z. Zhou for help in launching the simulations on the cluster.

## Author contributions

V.M.G. conceived, directed the project, participated to the experiments, supervised the experimental data analysis and interpretation. She supervised the simulations and analyzed the theoretical data. M.H. participated to the experiments, analyzed the experimental data and participated to the interpretation. H.L. thought and ran the finite element simulations and participated to the interpretation. A.T. and A.G. supervised the simulation code. F.C., D.D.A., D.F., L.F., R.M., E.Pa., E. Pe. and J.S.P.C. participated to the experiments. S.P. and F.B. partecipated to the experiments and in the discussion and interpretation of the data. V.M.G. wrote the manuscript, F.B., S.P.,F.C. and A.T. participated to the revisions.

## Competing interests

The authors declare no competing interests.

## Additional information

**Peer review information** : *Nature Communications* thanks the anonymous reviewer(s) for their contribution to the peer review of this work. A peer review file is available.

