## [Peer Review File · Nature Communications]

The effect of echoes interference on phonon attenuation in a nanophononic membraneReviewer #1 (Remarks to the Author):

Re: NCOMMS-23-37386-T

The effect of echoes interference on phonon dynamics in a nanophononic membrane
by Mohammad Hadi et al.

Referee Report

The paper is devoted to experimental measurements and theoretical modeling of phonon wavepackets mean free paths and phonon lifetimes in suspended uniform and nanostructured amorphous SiN membranes. It is shown that the nanostructuration results in the strong reduction in phonon lifetime, which is related in the paper with the contribution of coherent phonon interference. It is claimed that the coherent phonon interference becomes dominant for wavelengths comparable with the nanostructure lengthscale, the pitch and neck.

The statement of interference-induced phonon lifetime reduction in nanophononic membranes is interesting but it needs more deep explanation of its physical origin. For instance, the strong time decay of wavepacket kinetic energy, shown in figs. 4(a) and 6(a), can be explained by phonon scattering in a nanostructure of nanoholes with rough surfaces, without the contribution from the phonon interference. On the other hand, the not strictly periodic nanostructure with the proper average pitch of embedded nanoparticles as phonon scatterers can produce stop band through the phonon-interference resonance effects, see PRB 102, 024301 (2020) and references therein. Wavepacket propagation in the stop-band frequency/wavelength range results in the exponential spatial decay, which is caused by phonon interference in periodic structure.

In reviewer's opinion, the paper is interesting and can be published in Nature Communications upon the clarification of the physical origin of interference-induced phonon lifetime reduction.

Reviewer #2 (Remarks to the Author):

The manuscript studies coherent phonon interference in phononic crystal and its impact on phonon lifetime, which here is defined as time of mechanical oscillation fading. On the one hand, this is a good experiment and interesting analysis that show coherent phonon transport at somewhat surprisingly high frequencies for amorphous material. On the other hand, the findings and their interpretations are quite over-hyped. The manuscript claims everything "first time", with implications far beyond what the data really suggests. Besides that, I also need authors' help to see how measurements of phonon dispersion (which did not change in phononic crystal) are not at odds with all the claims of the manuscript. As a result, I invite the authors to revise the manuscript critically and specifically address some of the comments below:

1) First, I suggest improving the framing of this work in the abstract and introduction in the context of previous research. Maybe authors could better explain or correct these statements:

1.a) "The periodic nanostructuration strongly modifies phonon dispersions, allowing to effectively tailor thermal conductivity." - Actually, neither strong modifications of phonon dispersion nor thermal conductivity control were ever experimentally demonstrated at high enough frequencies/temperatures to make it realistically usable. So, I would not state it upfront as if it is well-known fact. There are some low frequency/temperature works like:

<https://pubs.acs.org/doi/full/10.1021/acs.nanolett.6b02305>

<https://www.science.org/doi/full/10.1126/sciadv.1700027>

<https://www.nature.com/articles/ncomms14054>

And they show that at room temperature, phonon dispersion is unlikely to change (at least above 100 GHz) and that no impact on the thermal conductivity exists due to such dispersion changes. There are some results on superlattices, but that is a bit different story.

1.b) "However, the role of periodicity in phonon attenuation remains unclear" - To me, the role of

periodicity seems to be quite clear actually. I know at least a few studies that investigate how periodicity impacts coherent modifications of heat conduction:

<https://www.nature.com/articles/ncomms4435>

<https://journals.aps.org/prb/abstract/10.1103/PhysRevB.93.045410>

<https://www.mdpi.com/2073-4352/6/6/72>

Longer periods lead to a stronger reduction of thermal conductivity. I understand that authors try to say that nobody studied the particular question of how quickly phonon waves decay in a phononic crystal, but there may be a better way to say that.

1.c) "The introduction of interfaces at the nanoscale is expected to affect the propagation of phonons with wavelength λ_{ph} comparable to the nanostructure length scale, reducing their lifetime τ and mean free path" - I don't think I agree with this statement. In the incoherent regime, this is correct, phonons are diffusely scattered by the interfaces and thus their lifetime is reduced. However, in the coherent (wave) regime, they scatter elastically (like waves) and preserve coherence and so their lifetime and free path continue. That is how they can develop phonon interference, which causes modification of the dispersion. Actually, the manuscript acknowledges this fact in the following sentences. So, maybe the authors could clarify this point. Again, I understand that the authors want to say that the lifetime, as defined in the manuscript, is affected, but maybe it can also be explained a bit better. It would be better if the lifetime was defined upfront in this manuscript.

1.d) "And indeed, a strong thermal conductivity reduction has been 80 reported [14]" - In this work, only about 20% reduction was observed in the very best case. Not really strong.

1.e) "Such wavelengths are not reachable with standard phonon spectroscopies such as inelastic X ray or neutron scattering, or Brillouin scattering" - I think Brillouin light scattering can reach such wavelengths and actually was used to demonstrate changes in phonon dispersion in nanophononic crystals:

<https://journals.aps.org/prb/abstract/10.1103/PhysRevB.91.075414>

Moreover, I know another paper where the coherence of phonons was probed in phononic crystals at these wavelengths:

<https://pubs.acs.org/doi/10.1021/acs.nanolett.6b02305>

2) I do not think SiN really has the properties that the manuscript assumes it has. For example: "SiN ... for which room temperature thermal transport has been reported to be dominated by phonons with $\ell \geq 1\mu\text{m}$ [36, 37]" - First, I don't see how [37] supports this statement. Second, I seriously doubt that in amorphous material MFP can be so long for any significant portion of phonons. If that were the case, its thermal conductivity would not be at the amorphous limit for all phononic structures except a few below 30 nm, as shown in [37].

And next: "And indeed, a thermal conductivity reduction of a factor up to 2 has been observed in nanoporous phononic SiN" - this reduction was barely observed for just three points below 30 nm, while all larger samples roughly showed values around the amorphous limit of 2.3 W/mK - not really a behavior of a material with MFP of one micron. So again, I don't see how this supports the statement of long MFP in SiN.

I mean, since the manuscript deals only with low-frequency phonons and not the thermal spectrum, I'd advise not to place these controversial statements about thermal properties as a premise of this work.

3) There are multiple places where the roughness is treated as a fitting parameter. I am not sure why it is so if the roughness was actually measured by AFM and it is not so far from the "fitted" value. So, I'd suggest just to fix it at the actually measured value.

4) I am not sure I understand the phrase "we have shown that the observed behavior is the result of a coherent mechanism which has never been discussed in such systems." - Surely, coherent scattering in phononic crystals leading to all kinds of localizations and attenuation has been widely discussed over the past 15 years. I think it has "never" been discussed in this exact way because

this definition of a lifetime as the time of oscillation amplitude fading is somewhat special. I can see how this definition might be acceptable, but all these "never" and "the first" are more like the first in this particular formulation of the problem. I recommend avoiding such statements. Again, in fact, the following paper did very similar experiments at similar wavelengths:
<https://pubs.acs.org/doi/10.1021/acs.nanolett.6b02305>

5) In the conclusion part "opening new perspectives in the physics of phonons and in the thermal engineering" - again, these frequencies really have very little to do with thermal engineering.

6) In the next conclusion: "Indeed, our data show how in nanophononic materials the nature of "phonons" at the nanoscale changes, becoming the superposition of coherent modes propagating in many different directions, as due to the reflections, and interfering with each other." - I really don't see how the "nature of phonons" is changing here. Yes, it's true that in some applications, we can consider phonons as particle-like wave-packet, but we are definitely aware that long-wavelength phonons can act as interfering waves.

7) I don't see how to link the data in Fig 2, which shows that the dispersion of phonons has essentially not changed at all, with other claims of the manuscript about coherent interference that changes everything, and in my understanding should change the dispersion like in a typical phononic crystal.

Reviewer #3 (Remarks to the Author):

Review report for the manuscript titled, "The effect of echoes interference on phonon dynamics in a nanophononic membrane" by M. Hadi et al.

The authors use extreme UV transient grating experiment to measure the phonon frequencies and lifetimes of phonons with 10's of GHz frequencies in bare and patterned amorphous thin SiN films. They find that the contribution of phonon interference effects are dominant in the patterned SiN films, resulting in a dramatic reduction in the observed phonon lifetimes compared to those in uniform films. They use finite element simulations to further validate their findings.

This work is interesting and the topic is probably suitable for Nature Communications. However, there are several technical questions that need to be addressed carefully in the manuscript, before I can discuss its suitability for Nat. Comm. further.

1. The authors use EUV source with wavelength ~ 10 's of nms. The thickness of the SiN membranes are ~ 55 nms. Have the authors considered finite penetration depth effects in their analysis?
2. As far as I know, the coherent effects can only affect the dispersions and group velocities. How do they affect the scattering rates? More clarity is needed on this point.
3. In fig. 2, there is hardly any difference in the frequencies of the uniform and the patterned membranes. How do the authors justify any coherent effects at all?
4. How do the authors quantify the roughness on the side walls of the pores in the patterned membranes?
5. Several previous works have cast doubts on the presence of any coherent effects on the phonons in lithographically nanopatterned thin films made of crystalline semiconductors, due to the possibility of strong phonon boundary scattering. Although the current work deals with amorphous films, the authors still invoke the picture of propagons (referring to Ref. [42] at a couple of places), implying that they believe the presence of plane-wave like energy carriers, which may also be susceptible to strong boundary scattering effects. In fact, this mechanism would directly result in the reduction of the lifetimes of these waves (in stark contrast with any coherent effects). Do the authors believe that significant reduction to the lifetimes of the energy carriers could have been caused by this effect also?
6. Similarly, a few works have shown that geometric effects such as back-reflection from the pore walls cause reduction in the contribution of phonons to the thermal conductivity of patterned crystalline nanomembranes. Can similar effects be possible for amorphous membranes as well?

7. The authors look at the effect of pores on the energy carriers with frequencies on the order of 100's of GHz. In the case of bulk amorphous SiN, what are the modes (with respect to frequencies) that contribute to thermal transport? In crystalline materials, phonons with frequencies on the order of a few THz are the primary heat carriers. If the same frequency range is also important for amorphous SiN, then perhaps, the authors should reconsider talking about the effects of holes/pores/patterns on the thermal conductivity of these films.

8. A minor point: The caption to Fig. 8 talks about a blue solid line, but I don't see one in the figure. I request the authors to fix this inconsistency.

REVIEWER COMMENTS

Reviewer #1 (Remarks to the Author):

Re:NCOMMS-23-37386-T

The effect of echoes interference on phonon dynamics in a nanophononic membrane by Mohammad Hadi et al.

Referee Report

The paper is devoted to experimental measurements and theoretical modeling of phonon wavepackets mean free paths and phonon lifetimes in suspended uniform and nanostructured amorphous SiN membranes. It is shown that the nanostructuration results in the strong reduction in phonon lifetime, which is related in the paper with the contribution of coherent phonon interference. It is claimed that the coherent phonon interference becomes dominant for wavelengths comparable with the nanostructure lengthscale, the pitch and neck.

The statement of interference-induced phonon lifetime reduction in nanophononic membranes is interesting but it needs more deep explanation of its physical origin. For instance, the strong time decay of wavepacket kinetic energy, shown in figs. 4(a) and 6(a), can be explained by phonon scattering in a nanostructure of nanoholes with rough surfaces, without the contribution from the phonon interference. On the other hand, the not strictly periodic nanostructure with the proper average pitch of embedded nanoparticles as phonon scatterers can produce stop band through the phonon-interference resonance effects, see PRB 102, 024301 (2020) and references therein. Wavepacket propagation in the stop-band frequency/wavelength range results in the exponential spatial decay, which is caused by phonon interference in periodic structure.

In reviewer's opinion, the paper is interesting and can be published in Nature Communications upon the clarification of the physical origin of interference-induced phonon lifetime reduction.

We thank the Referee for his/her comments and for posing the crucial question about the origin of this interference-related phonon lifetime reduction

The incoherent boundary scattering due to roughness in the experimental data is taken into account through the τ_b term, which is considered as a free parameter and therefore also includes (empirically) the roughness of the holes. However, this term cannot explain the observed time decay, neither if an additional channel due to anharmonicity is considered. The additional lifetime simulated by finite element modelling (FEM), τ_{coh} , is not related to roughness, since the calculations assume perfectly flat interfaces, nor to anharmonicity, which is also not considered in the calculations. It thus accounts for the decay of acoustic modes (propagating along a given direction) due to coherent phonon scattering and interference in the nanophononic structure. This is actually the main result of the present work.

Concerning the presence of stop bands, we have run phonon dispersion calculations at the experimental frequencies and we did not find any stop band. We thank the referee for the suggestion of the PRB paper, which we did not know. This paper shows a destructive interference between the phonons from the matrix and local resonances from the nanoparticles, which indeed reminds of the effects reported in some phononic crystals. However, we don't think we are in a similar case, since the two-paths stop band mechanism needs local resonances to interfere with the phonons of the matrix, while we have empty holes, which makes resonant effects very unlikely. Moreover, in the manuscript mentioned by the referee the nanoparticles are much smaller than the acoustic wavelength, while the lengthscales (period, neck and pores diameter) of our nanostructure are larger than/comparable to the acoustic wavelength.

The fact that the mean free path of the wave-packet becomes shorter in correspondence of stronger interference in the kinetic energy at a given position indicates that the attenuation is related to a coherent mechanism. The interference arises from wave-packets reflected off the circular pores in many directions, almost over 2π . Reflected wave-packets can interfere with each other at later times (like "echoes"), since they do not follow a straight propagation and therefore arrive later (or come back) at a given position. In addition, if the coherence length of the wave-packets is longer than the path difference between the main wave-packet (i.e. the one that is not-reflected at all and thus propagates in the straight direction – x-axis) and the reflected ones, they can also interfere. This is the case of wave-packets reflected in near forward directions, which can interfere with the later time tail of the main wave-packet before it has completely moved on. For given path length differences between two wave-packets, interference can be constructive or destructive, thus enhancing or depleting the amplitude of the interference pattern. In our case, we have a broad distribution of wave-packets, coming from different angles and with different path length differences, which can interfere and thus redistribute the amplitude of the vibrational mode in time and space. This is exactly what is calculated and shown in Figures 4 and 6. We have added this explanation in the manuscript (page 4/5).

Reviewer #2 (Remarks to the Author):

The manuscript studies coherent phonon interference in phononic crystal and its impact on phonon lifetime, which here is defined as time of mechanical oscillation fading. On the one hand, this is a good experiment and interesting analysis that show coherent phonon transport at somewhat surprisingly high frequencies for amorphous material. On the other hand, the findings and their interpretations are quite over-hyped. The manuscript claims everything "first time", with implications far beyond what the data really suggests. Besides that, I also need authors' help to see how measurements of phonon dispersion (which did not change in phononic crystal) are not at odds with all the claims of the manuscript. As a result, I invite the authors to revise the manuscript critically and specifically address some of the comments below:

1) First, I suggest improving the framing of this work in the abstract and introduction in the context of previous research. Maybe authors could better explain or correct these statements:

1.a) "The periodic nanostructuring strongly modifies phonon dispersions, allowing to effectively tailor thermal conductivity." - Actually, neither strong modifications of phonon dispersion nor thermal conductivity control were ever experimentally demonstrated at high enough frequencies/temperatures to make it realistically usable. So, I would not state it upfront as if it is well-known fact. There are some low frequency/temperature works like: <https://pubs.acs.org/doi/full/10.1021/acs.nanolett.6b02305>
<https://www.science.org/doi/full/10.1126/sciadv.1700027>
<https://www.nature.com/articles/ncomms14054>

And they show that at room temperature, phonon dispersion is unlikely to change (at least above 100 GHz) and that no impact on the thermal conductivity exists due to such dispersion changes. There are some results on superlattices, but that is a bit different story.

We do thank the Referee for his/her critical reading and suggestions, as well as for the suggested references (we added some of them in the revised version of the manuscript). First of all, we tried to remove the "over-hype" throughout the paper.

Concerning this specific point, we agree that we should have been more precise on the role of nanostructuring for tailoring thermal transport. The sentence mentioned by the Referee generally refers to nanophononic materials, thus including super-lattices, and not only the material of this study. Indeed, this was not clear, therefore we decided to remove this sentence and rewrote the abstract accordingly.

The works cited by the Referee show tangible effects only at very low temperature and low frequency (< 100 GHz). In the papers by Wagner *et al.* and Maire *et al.* the relatively large roughness (7 nm for the former and 3-4 nm for the latter) likely prevents to observe any effect on thermal transport at room temperature. Indeed, it is dominated by phonons in the THz range, i.e. with wavelengths below 10 nm, as correctly mentioned by the referee. In the revised version of the manuscript we stressed more the crucial role of the low roughness (< 1 nm) of our structure. In the third paper, Lee *et al.* conclude that: "...the wave nature of phonons does not need to be considered to describe transport in the regime where $\lambda \ll p \ll \Lambda_U$ and $\lambda \sim \delta$ ", i.e. for phonon wavelengths (λ) much smaller than the periodicities (p) but comparable to the roughness (δ), and periodicities much smaller than the mean free path due to Umklapp scattering (Λ_U). This is not the present case, since $\lambda \gg \delta$. The low roughness is an essential

ingredient, which is very difficult to achieve for phonons which usually dominate heat transport at room temperature, such as the ones with THz frequencies and wavelengths in the nanometer/sub-nanometer range.

However, the interest of nanoporous membranes for thermal transport engineering is of relevance, as also shown in the recent review by Nomura et al., *Materials Today Physics* 22, 100613 (2022). Indeed, while the contribution to thermal transport of high frequency phonons can be efficiently reduced by doping or anharmonicity, manipulation of the lower frequency phonons can be considered the key for further suppression of the thermal conductivity (10.1103/PhysRevB.102.024301). As an example, a coherent contribution to the thermal conductivity reduction at room temperature in nanophononic porous materials was reported in Refs [14] and [15] of our manuscript. Our work essentially suggests how manipulation of phonons in the 100 GHz range can be effectively achieved via coherent effects (i.e. phonon interference) induced by the nanostructure itself.

1.b) "However, the role of periodicity in phonon attenuation remains unclear" - To me, the role of periodicity seems to be quite clear actually. I know at least a few studies that investigate how periodicity impacts coherent modifications of heat conduction: <https://www.nature.com/articles/ncomms4435>
<https://journals.aps.org/prb/abstract/10.1103/PhysRevB.93.045410>
<https://www.mdpi.com/2073-4352/6/6/72>

Longer periods lead to a stronger reduction of thermal conductivity. I understand that authors try to say that nobody studied the particular question of how quickly phonon waves decay in a phononic crystal, but there may be a better way to say that.

The Referee is right, the meaning of our sentence is exactly: "*nobody studied the particular question of how quickly phonon waves decay in a phononic crystal*". To address this comment and the one in point 1a, we have modified the abstract to moderate the claim and to make these points clearer.

1.c) "The introduction of interfaces at the nanoscale is expected to affect the propagation of phonons with wavelength λ_{ph} comparable to the nanostructure length scale, reducing their lifetime τ and mean free path" - I don't think I agree with this statement. In the incoherent regime, this is correct, phonons are diffusely scattered by the interfaces and thus their lifetime is reduced. However, in the coherent (wave) regime, they scatter elastically (like waves) and preserve coherence and so their lifetime and free path continue. That is how they can develop phonon interference, which causes modification of the dispersion. Actually, the manuscript acknowledges this fact in the following sentences. So, maybe the authors could clarify this point. Again, I understand that the authors want to say that the lifetime, as defined in the manuscript, is affected, but maybe it can also be explained a bit better. It would be better if the lifetime was defined upfront in this manuscript.

We thank the Referee for pointing out our lack of clarity. We removed this sentence and modified the following ones in order to be more clear.

1.d) "And indeed, a strong thermal conductivity reduction has been 80 reported [14]" - In this work, only about 20% reduction was observed in the very best case.

We agree with the referee, indeed, it is the wrong reference. The correct one is ref. [14] in our manuscript. We have corrected this mistake.

1.e) "Such wavelengths are not reachable with standard phonon spectroscopies such as inelastic X ray or neutron scattering, or Brillouin scattering" - I think Brillouin light scattering can reach such wavelengths and actually was used to demonstrate changes in phonon dispersion in nanophononic crystals:

<https://journals.aps.org/prb/abstract/10.1103/PhysRevB.91.075414>

Moreover, I know another paper where the coherence of phonons was probed in phononic crystals at these wavelengths:

<https://pubs.acs.org/doi/10.1021/acs.nanolett.6b02305>

Indeed, Brillouin backscattering of visible light can allow reaching wavelengths in the 250 nm range, as in the first paper mentioned by the referee. However, many nanostructures designed for tailoring thermal transport feature a smaller lengthscale (a few tens of nanometers), which escape Brillouin scattering but remains still too large for inelastic neutron and x-ray scattering techniques. Furthermore, inelastic scattering experiments in general suffer from the reduced scattering volume due to very thin samples. Pump-probe approaches, like the one used in the second manuscript mentioned by the referee, can probe phonons travelling perpendicular to the surface of the membrane, while the ones that we are looking for are those travelling parallel to it. These can be probed by the transient grating technique, which, however, is limited to $\approx 1 \mu\text{m}$ wavelengths when optical pulses are used. The use of EUV light removes this limitation and potentially allows reaching wavelengths down to $\approx 10 \text{ nm}$. We reformulated this sentence to point out the correct range of Brillouin light scattering and we also mention picosecond ultrasonics approaches (page 1)

2) I do not think SiN really has the properties that the manuscript assumes it has. For example: "SiN ... for which room temperature thermal transport has been reported to be dominated by phonons with $\ell \geq 1 \mu\text{m}$ [36, 37]" - First, I don't see how [37] supports this statement. Second, I seriously doubt that in amorphous material MFP can be so long for any significant portion of phonons. If that were the case, its thermal conductivity would not be at the amorphous limit for all phononic structures except a few below 30 nm, as shown in [37].

And next: "And indeed, a thermal conductivity reduction of a factor up to 2 has been observed in nanoporous phononic SiN" - this reduction was barely observed for just three points below 30 nm, while all larger samples roughly showed values around the amorphous limit of 2.3 W/mK - not really a behavior of a material with MFP of one micron. So again, I don't see how this supports the statement of long MFP in SiN.

I mean, since the manuscript deals only with low-frequency phonons and not the thermal spectrum, I'd advise not to place these controversial statements about thermal properties as a premise of this work.

Following the suggestions of the Referee, we have modified the sentence at page 2 and taken out the controversial statements on thermal transport, only mentioning the argument that phonons in the investigated wavelength and frequency range are predicted to have a mean free path in the micron range, that, actually, our measurements confirm (J. Appl. Phys. 130, 035101 (2021)).

However, we mention that such long mean free paths are not unusual in amorphous materials in the 100 GHz frequency range. Indeed, disordered materials usually feature phonon mean free path of the order of few nanometers for phonons with wavelengths in the nanometer/sub-nanometer range, while the phonon mean free path can be much longer at smaller frequencies/longer wavelengths. As can be seen in the figure below, extracted from Phys. Rev. Mat. 5, 065602 (2021), phonons in amorphous Si and SiO₂ have mean free paths in the order of 1-50 microns at frequencies of 100 GHz, comparable to ours.

FIG. 4. Reconstructed MFPs versus frequency for thermal acoustic excitations in aSi at 300 K. Also shown are PSA data for aSi (Ref. [37]), literature data for vitreous silica from inelastic x-ray scattering (diamonds, Refs. [8,55]), tunnel junction spectroscopy (5-pointed stars, Ref. [14]), a multipulse optical technique (upward pointing triangles, Ref. [39]), and from transport measurements (dashed line, Ref. [31]).

3) There are multiple places where the roughness is treated as a fitting parameter. I am not sure why it is so if the roughness was actually measured by AFM and it is not so far from the "fitted" value. So, I'd suggest just to fix it at the actually measured value.

We measured the roughness of the upper surface only. We do not have the certainty that all surfaces have the same roughness. The value obtained by the fit (0.6-0.7 nm) is close to the measured one (0.4 nm) and, furthermore, it is reasonable to assume that the top surface is the flatter one. Therefore, an effective roughness somewhat larger than the one measured in the top surface is not surprising. We have added this comment in the Methods section.

4) I am not sure I understand the phrase "we have shown that the observed behavior is the result of a coherent mechanism which has never been discussed in such systems." - Surely, coherent scattering in phononic crystals leading to all kinds of localizations and attenuation has been widely discussed over the past 15 years. I think it has "never" been discussed in this exact way because this definition of a lifetime as the time of oscillation amplitude fading is somewhat special. I can see how this definition might be acceptable, but all these "never" and "the first" are more like the first in this particular formulation of the problem. I recommend avoiding such

statements. Again, in fact, the following paper did very similar experiments at similar wavelengths:

<https://pubs.acs.org/doi/10.1021/acs.nanolett.6b02305>

We agree that coherent phonons in phononic crystals have been widely discussed. However, as far as we know, experimental findings on the effects of the coherent mechanism of phonon wave interference on the attenuation of a given acoustic mode at such length scales have not yet been reported. The paper cited by the Referee does not analyse phonon attenuation, furthermore, they measured phonons in the direction perpendicular to the surface of the membrane, while we measured phonons parallel to the surface, which we expect to be the ones more strongly affected by nanostructuring.

We also think that the definition of phonon attenuation as the fading away of the oscillations in the TG signal is not special. In continuous media this decay can be associated, for instance, to the phonon lifetime entering the Boltzmann transport equation in the single relaxation time approximation. Here we apply this definition to interpret the signal from the nanostructure. Finite element calculations allow us to see that a stronger reduction of the mean free path of the wave-packet takes place in correspondence of stronger interference effects in the kinetic energy at a given position, thus suggesting that the attenuation is related to a coherent mechanism. The interference arises from wave-packets reflected off the pores in many directions, since the pores are circular. Reflected wave-packets can interfere with each other at later times (like “echoes”), since they do not follow a straight propagation and therefore arrive later at a given position. In addition, if the coherence length of the wave-packets is longer than the path difference between the main wave-packet (i.e. the one that is not-reflected at all and thus propagates in the straight direction – x-axis) and the reflected ones, they can also interfere. This is the case of wave-packets reflected in near forward directions, which can interfere with the later time tail of the main wave-packet before it has completely moved on. For given path length differences between two wave-packets, interference can be constructive or destructive, thus enhancing or depleting the amplitude of the interference pattern. In our case, we have a broad distribution of wave-packets, coming from different angles and with different path length differences, which can interfere and thus redistribute the amplitude of the vibrational mode in time and space. This is exactly what is calculated and shown in Figure 4 and 6. We have added this explanation in the manuscript (page 4/5).

We have mitigated our claims and better explained our results in page 6.

5) In the conclusion part "opening new perspectives in the physics of phonons and in the thermal engineering" - again, these frequencies really have very little to do with thermal engineering.

We have smoothed down our claim in the conclusions.

6) In the next conclusion: "Indeed, our data show how in nanophononic materials the nature of "phonons" at the nanoscale changes, becoming the superposition of coherent modes propagating in many different directions, as due to the reflections, and interfering with each other." - I really don't see how the "nature of phonons" is changing here. Yes, it's true that in some applications, we can consider phonons as particle-like wave-packet, but we are definitely aware that long-wavelength phonons can act as interfering waves.

We wanted to say that reflections and interferences induce a spreading of the mechanical energy impulsively generated by the TG in many directions. We have changed this sentence to make our point clearer in the conclusions (page 6).

7) I don't see how to link the data in Fig 2, which shows that the dispersion of phonons has essentially not changed at all, with other claims of the manuscript about coherent interference that changes everything, and in my understanding should change the dispersion like in a typical phononic crystal.

In the nanophononic membrane we expect phonon dispersions to be modified by the zone folding effect. In Fig.2 we have compared our experimental frequencies with the analytical calculation of Lamb dispersions for a uniform (non nanopatterned) membrane with a reduced density and Young modulus. The good agreement is indeed surprising, but, as we point out in the manuscript, not all of our fitted acoustic modes can be reproduced, suggesting that they come from the zone folding effect not taken into account in the simple model. Such additional modes do suggest a modification of branches due to the nanophononic structure. In fact, also at the other experimental wavelengths the fit could be improved by adding more phonon modes, at frequencies at which the Fourier Transform presents some minor features. The modes reported in the manuscript are the ones which give the stronger signal, meaning that they have the largest cross section (they are typically those having frequencies closer to the longitudinal acoustic one; see Ref. [41]). We have chosen to include the minimum possible number of modes in our fit, in order to keep the number of free parameters as low as possible. We have added a sentence on this in the revised manuscript on page 3.

Reviewer #3 (Remarks to the Author):

Review report for the manuscript titled, "The effect of echoes interference on phonon dynamics in a nanophononic membrane" by M. Hadi et al.

The authors use extreme UV transient grating experiment to measure the phonon frequencies and lifetimes of phonons with 10's of GHz frequencies in bare and patterned amorphous thin SiN films. They find that the contribution of phonon interference effects are dominant in the patterned SiN films, resulting in a dramatic reduction in the observed phonon lifetimes compared to those in uniform films. They use finite element simulations to further validate their findings.

This work is interesting and the topic is probably suitable for Nature Communications. However, there are several technical questions that need to be addressed carefully in the manuscript, before I can discuss its suitability for Nat. Comm. further.

We thank the Referee for his/her critical reading of our manuscript and here in the following we answer to the raised concerns.

1. The authors use EUV source with wavelength ~ 10 's of nms. The thickness of the SiN membranes are ~ 55 nms. Have the authors considered finite penetration depth effects in their analysis?

As reported in Table 1, the pump penetration depth is 54, 26 and 10 nm (from the lowest to the largest excitation wavelength), while the probe penetration length is always larger than the membrane thickness. As such, we probe a region larger than the one excited by the pump. Still, only in the region where the pump is absorbed there will be the generation of acoustic modes, that are detected by transient diffraction. A signature of finite penetration depth effects in TG experiments on thin samples is the detection of asymmetric Lamb modes (Journal of Applied Physics 67, 3362 (1990)), which we do not see, so we disregarded this aspect. The finite penetration depth also affects the EUV TG signal intensity, as explained in Ref [25], but this is not relevant in the present context, since we do not discuss signal intensity at all. We have added a sentence on this point in the Methods section.

Concerning the thermal relaxation, which contributes with an exponential term to the signal (see Eq. 1 in the paper), the situation is different: here we cannot safely assume a 1D thermal transport (only from hot to cold fringes), as there could be an important thermal transport from the uppermost (hot) layer of the membrane to the lower (cold) one. Despite this, we observe a diffusive thermal transport in the uniform membrane, as already reported [see Refs. 40-41]. In the nanophononic membrane we observe a deviation from the diffusive transport at our longest wavelength, corresponding to the shortest penetration depth. This deviation could indeed be related to a not-unidimensional thermal transport, which might be more prominent in the nanophononic membrane than in the uniform one. In order to address this interesting aspect more data are needed, as already mentioned in the SM.

2. As far as I know, the coherent effects can only affect the dispersions and group velocities. How do they affect the scattering rates? More clarity is needed on this point.

This is indeed the key point of our paper, the question we try to answer is: “Is there an effect on scattering rate?” According to the present results, we do see a stronger attenuation of the modes travelling along the x-axis through the nanophononic structure, when interference becomes more important. The fact that the mean free path of the wave-packet becomes shorter in correspondence of stronger interference in the kinetic energy at a given position indicates that the attenuation is related to a coherent mechanism. The interference arises from wave-packets reflected off the circular pores in many directions, from near backward to near forward. Reflected wave-packets can interfere with each other at later times (like “echoes”), since they do not follow a straight propagation and therefore arrive later at a given position. In addition, if the coherence length of the wave-packets is longer than the path difference between the main wave-packet (i.e. the one that is not-reflected at all and thus propagates in the straight direction – x-axis) and the reflected ones, they can also interfere. This is the case of wave-packets reflected in near forward directions, which can interfere with the later time tail of the main wave-packet before it has completely moved on. For given path length differences between two wave-packets, interference can be constructive or destructive, thus enhancing or depleting the amplitude of the interference pattern. In our case, we have a broad distribution of wave-packets, coming from different angles and with different path length differences, which can interfere and thus redistribute the amplitude of the vibrational mode in time and space. This is exactly what is calculated and shown in Figure 4 and 6. We have added this explanation in the manuscript (page 4/5).

3. In fig. 2, there is hardly any difference in the frequencies of the uniform and the patterned membranes. How do the authors justify any coherent effects at all?

In the nanophononic membrane we expect phonon dispersions to be modified by the zone folding effect. In Fig.2 we have compared our experimental frequencies with the analytical calculation of Lamb dispersions for a uniform (non nanopatterned) membrane with a reduced density and Young modulus. The good agreement is indeed surprising, but, as we point out in the manuscript, not all of our fitted phonon modes can be reproduced: there are 3 extra-modes at the shortest wavelength in the nanostructured sample, absent in the uniform one. These modes likely come from the zone folding effect not taken into account in the simple model. Such additional modes do suggest a modification of branches due to the nanophononic structure. In fact, also at the other experimental wavelengths the fit could be improved by adding more phonon modes, at frequencies at which the Fourier Transform presents some minor features. Still, the modes reported in the manuscript are the ones which give the major signal, meaning that they have the largest cross section (they are typically those having frequencies closer to the longitudinal acoustic one; see Ref. [41]). We have chosen to include in our fit the minimum possible number of modes, in order to keep the number of free parameters as low as possible. We have added a sentence on this in the revised manuscript at page 3.

4. How do the authors quantify the roughness on the side walls of the pores in the patterned membranes?

This roughness is taken into account in the τ_{BD} term and fitted as a free parameter. The fitting result can be considered as an effective roughness, which empirically includes the contribution of top and bottom surfaces, as well as the walls of the pores, similarly to what proposed by Maire *et al.*, Sc. Adv. 2017 (doi: 10.1126/sciadv.1700027). Additionally, we only measured the roughness of the upper surface. We do not have the certainty that all surfaces have

the same roughness, however, the value of the effective roughness obtained by the fit (0.6-0.7 nm) is close to the measured one (0.4 nm). It is reasonable to assume that the top surface is the flattest and, therefore, an effective roughness somewhat larger than the one measured in the top surface is not surprising. We have added this comment in the Methods section.

5. Several previous works have cast doubts on the presence of any coherent effects on the phonons in lithographically nanopatterned thin films made of crystalline semiconductors, due to the possibility of strong phonon boundary scattering. Although the current work deals with amorphous films, the authors still invoke the picture of propagons (referring to Ref. [42] at a couple of places), implying that they believe the presence of plane-wave like energy carriers, which may also be susceptible to strong boundary scattering effects. In fact, this mechanism would directly result in the reduction of the lifetimes of these waves (in stark contrast with any coherent effects). Do the authors believe that significant reduction to the lifetimes of the energy carriers could have been caused by this effect also?

We do consider boundary scattering in the τ_{BD} term in our model already in the uniform membrane. This is modelled with the Ziman formulation for boundary scattering in presence of roughness. This boundary scattering is slightly more important in the nanophononic membrane due to the presence of the walls with their own roughness, but the experimental lifetimes as a function of the mode's wavelength cannot be reproduced only with boundary scattering, nor when an additional term (anharmonicity) is added.

6. Similarly, a few works have shown that geometric effects such as back-reflection from the pore walls cause reduction in the contribution of phonons to the thermal conductivity of patterned crystalline nanomembranes. Can similar effects be possible for amorphous membranes as well?

We do believe so, and we suggest that this is what happens at the shortest wavelengths, where we see in the FEM simulations a first signal coming from near-backward reflections, but not later interference signals appearing (see SM, Fig. 7 and 8 and discussion page 10/11).

7. The authors look at the effect of pores on the energy carriers with frequencies on the order of 100's of GHz. In the case of bulk amorphous SiN, what are the modes (with respect to frequencies) that contribute to thermal transport? In crystalline materials, phonons with frequencies on the order of a few THz are the primary heat carriers. If the same frequency range is also important for amorphous SiN, then perhaps, the authors should reconsider talking about the effects of holes/pores/patterns on the thermal conductivity of these films.

Our study aims at understanding how phonons are actually affected by the nanostructure, beyond the material actually used for the experiment. We find a strong phonon attenuation due to coherent interference and this effect is the strongest at wavelengths similar to the neck. This indicates that, for affecting thermal transport in an effective way, we need to engineer the nanostructure so that the neck is close to the wavelength of the phonon modes that are primary heat carriers for a given material and working conditions. While this result could be expected, our findings concern how it happens. We have reformulated the interest for thermal transport in the whole paper.

Just to answer the question of the Referee about thermal transport in SiN, recent calculations, reported in *J. Appl. Phys.* 130, 035101 (2021), have reported on the role of propagative phonons (propagons) in thermal transport in a-SiN. They show that propagons contribute to 70% of the total thermal conductivity at room temperature and their attenuation is mostly due to the Akhiezer anharmonic mechanism. From the reported lifetimes, a mean free path of about 75 nm in the THz range, and 6 to 8 microns at our experimental frequencies is calculated, quite in agreement with our estimated mean free path in the uniform membrane, which is lower, as expected, as due to the boundary scattering.

On the experimental side, a recent work, *ACS Nano* 2020, 14, 6, 6980–6989, reports on the reduction of thermal conductivity in a nanophononic membrane of SiN, of only 8% for a periodicity similar to ours, but of as much as 60% for a smaller periodicity, as also reported in *Science Advances* 6, 750 (2020).

8. A minor point: The caption to Fig. 8 talks about a blue solid line, but I don't see one in the figure. I request the authors to fix this inconsistency.

Indeed, there is only the blue dashed line, this is a mistake. We thank the Referee for pointing it out.

Reviewer #1 (Remarks to the Author):

Re: NCOMMS-23-37386-T

The effect of echoes interference on phonon dynamics in a nanophononic membrane
by Mohammad Hadi et al.

Second Referee Report

In the revised version of the paper and in the answers to all the reviewers, the authors explained in additional words why they relate the phonon lifetime reduction in a low-roughness nanoporous phononic membrane of SiN with the coherent phonon-interference effects.

Although these additional explanations are a bit conjectural, in reviewer's opinion the paper presents interesting experimental and numerical results on the dependence of phonon frequencies and lifetimes on the nanostructuring of amorphous membranes and can be published in Nature Communications.

Reviewer #2 (Remarks to the Author):

I think most of my comments were addressed and the manuscript has been improved so I can recommend publication.

Reviewer #3 (Remarks to the Author):

I would like to thank the authors for taking the time to address my concerns. After reading their responses to my comments as well as those of the other reviewers. I am still not convinced that there is sufficient new physics in the manuscript that warrants a publication in Nature Communications.

With respect to my first question, it appears that the authors have not taken into account the finite penetration depth effect into account carefully. I would think that the finite penetration effects will play a dominant role in thermal transport at TG wavelengths longer than ~ 20 nm (based on the optical penetration depths provided by the authors in the response), just because the pump wouldn't heat up the sample beyond ~ 20 nm thickness sufficiently and so, there will be significant cross-plane heat transport as well. Hence, further careful analysis is needed in this regard to ascertain the conclusions of this manuscript.

Furthermore, overall I believe that the coherent effects that I'm familiar with, is different from what has been portrayed in this manuscript. Coherent effects arises out of wave interference, which must necessarily modify the phonon dispersions and group velocities. These effects might modify the scattering rates as well, but predominantly through the scattering phase space in the material. I don't see a clear direct reason for the anharmonicity to be modified by wave interference. However, from what I understand, the authors seem to equate back-scattering effects from the nanoscale holes in the sample to coherent effects. This concern seems to be consistent with some of the other reviewers' concerns as well. Additionally, statements such as "The fact that the mean free path of the wave-packet becomes shorter in correspondence of stronger interference in the kinetic energy at a given position indicates that the attenuation is related to a coherent mechanism" in the response report make it more confusing for me, since I don't quite follow the role of the "kinetic energy at a given position" here.

Finally, it is not clear if the back-scattering is a coherent specular reflection either (since the roughness of the side walls of the holes can't be measured). These reflections could just be incoherent, phase-destroying diffuse scattering off the pores, which cause significant resistance to heat flow and cannot realistically be modeled by a single adjustable τ_{BD} parameter, as far as I know. The observation that incoherent back scattering of phonons cause additional thermal resistance in nanoscale structures has been done previously by several research groups in the

past.

Hence, I do not think this manuscript is suitable for Nature Communications, but rather more suitable to a specialized journal, where the developed experimental technique and the confirmatory results will be of more direct interest.

REVIEWER COMMENTS

Reviewer #1 (Remarks to the Author):

Re: NCOMMS-23-37386-T

The effect of echoes interference on phonon dynamics in a nanophononic membrane by Mohammad Hadi et al.

Second Referee Report

In the revised version of the paper and in the answers to all the reviewers, the authors explained in additional words why they relate the phonon lifetime reduction in a low-roughness nanoporous phononic membrane of SiN with the coherent phonon-interference effects.

Although these additional explanations are a bit conjectural, in reviewer's opinion the paper presents interesting experimental and numerical results on the dependence of phonon frequencies and lifetimes on the nanostructuring of amorphous membranes and can be published in Nature Communications.

Reviewer #2 (Remarks to the Author):

I think most of my comments were addressed and the manuscript has been improved so I can recommend publication.

We thank Reviewers 1 and 2 for their positive feedbacks.

Reviewer #3 (Remarks to the Author):

I would like to thank the authors for taking the time to address my concerns. After reading their responses to my comments as well as those of the other reviewers. I am still not convinced that there is sufficient new physics in the manuscript that warrants a publication in Nature Communications.

With respect to my first question, it appears that the authors have not taken into account the finite penetration depth effect into account carefully. I would think that the finite penetration effects will play a dominant role in thermal transport at TG wavelengths longer than ~ 20 nm (based on the optical penetration depths provided by the authors in the response), just because the pump wouldn't heat up the sample beyond ~ 20 nm thickness sufficiently and so, there will be significant cross-plane heat transport as well. Hence, further careful analysis is needed in this regard to ascertain the conclusions of this manuscript.

As already said in our previous answer and stated in the manuscript, we agree that the finite penetration depth of the pump can alter cross-plane heat transport. We also acknowledged the possibility that cross-plane heat transport can affect the in-plane thermoelastic dynamics (we remark that in this experiment we are sensitive to in-plane thermoelastic dynamics). Indeed, we do comment in the supplementary material section about the indication that the thermal relaxation of the TG is affected by the finite penetration depth of the pump. The data point at $q=0.057$ nm⁻¹ in Fig S3 of the supplementary material, corresponding to the shorter pump's penetration depth, seems out of trend and we already mentioned that this could be due to a

“cross-talk” between in-plane and cross-plane heat transport. However, with the data in hand, we cannot draw any conclusion on this interesting aspect; further studies are needed to assess the nanoscale thermal relaxation dynamics in these membranes. Concerning phonon dynamics, in TG experiments a signature of tangible effects of the finite penetration depth is the excitation and detection of antisymmetric Lamb modes. Since we do not observe prominent antisymmetric modes, we excluded large effects on phonon dynamics, in particular in the large decrease of phonon lifetime in the nanostructured sample vs the uniform membrane; we stress that the penetration depth in SiN does not change besides a marginal effect due to a slight change in SiN density. We hereby recall that such a reduction in phonon lifetime is the main result of the present work. We do believe that this reasoning is sufficient to address this remark. While it was already outlined in the previously revised version at page 7, we modified the manuscript to make a more proper reference to the supplementary material section in order to improve clarity.

Another aspect related to this remark is a possible wrong estimation of the temperature raise due to the pump. This enters in our data analysis through the value of τ_{ph-ph} ; see Eq.2. We have checked that assuming different temperatures up to 100 C, it does not affect significantly the fitting results: it only leads to a slightly different value of the anharmonic parameter B. Most importantly, this does not change the physical interpretation of the role of coherent mechanisms on the phonon lifetime in the nanostructured membrane, since the dependence of τ on ν cannot be explained if the τ_{coh} term is ignored. We have added this consideration in the Methods section when we comment on the calculated temperature raise.

Furthermore, overall I believe that the coherent effects that I’m familiar with, is different from what has been portrayed in this manuscript. Coherent effects arises out of wave interference, which must necessarily modify the phonon dispersions and group velocities. These effects might modify the scattering rates as well, but predominantly through the scattering phase space in the material. I don’t see a clear direct reason for the anharmonicity to be modified by wave interference. However, from what I understand, the authors seem to equate back-scattering effects from the nanoscale holes in the sample to coherent effects. This concern seems to be consistent with some of the other reviewers’ concerns as well. Additionally, statements such as “The fact that the mean free path of the wave-packet becomes shorter in correspondence of stronger interference in the kinetic energy at a given position indicates that the attenuation is related to a coherent mechanism” in the response report make it more confusing for me, since I don’t quite follow the role of the “kinetic energy at a given position” here.

We apologize if we haven’t explained this crucial point clearly. We did not claim that wave interference substantially modifies anharmonicity, indeed, we found that the anharmonic parameter B slightly increases (from 1.7 ± 0.6 to $2.7 \pm 0.8 \text{ m}^3\text{s}^{-2}\text{K}^{-1}$), which can be qualitatively explained in terms of additional channels for phonon-phonon scattering in the nanostructure, due to new low-lying optic modes arising from branch folding (i.e. the modification of the scattering phase space cited by the Referee). However, this is not the key point of our work. The key point is that we have identified another phonon decay mechanism arising from the interference of the original phonon wave-packet with its own reflections at the interfaces. This mechanism was identified by finite elements calculations, performed on an ideal material, with no roughness and no anharmonicity, in order to unambiguously evidence coherent effects coming only from the geometry of the nanostructure. This is the term τ_{coh} , which in Eq. 2 we empirically add to the other two phonon decay mechanisms, i.e. surface roughness (τ_b) and anharmonicity (τ_{ph-ph}). Only when this additional term is considered the experimentally

observed dependence of τ on ν can be explained. We do firmly believe that this is a proof of the existence of a coherent effect on phonon decay, beyond the modification of the scattering phase space for anharmonicity. We have added this comment in the Supplementary Material.

We don't equate backscattering with coherent effects. As stated above and as written in the manuscript, calculations do not consider roughness, therefore the wavepacket is coherently scattered by holes. In the real material, there is also incoherent scattering due to roughness, this is empirically accounted for by the τ_b term in Eq. 2. The indication that the effective roughness in the holey sample (0.7 ± 0.1 nm) is larger than the one (0.6 ± 0.1 nm) in the uniform membrane is qualitative consistent with the expectation that the holes are somewhat rougher than the membrane itself. In short, the τ_{coh} term identified by calculations can actually be regarded as the decay in an ideal nanostructure, with no roughness nor anharmonic interactions.

Concerning the instantaneous kinetic energy (E_{kin}): this is the vibrational energy summed on the cross section along the propagation direction (x-axis), i.e. the total vibrational energy in the (y,z)-plane for a given value of x (see Fig. 4a for the reference frame). As we have created a pulse travelling along the x-axis and we do not consider attenuation mechanisms, in the uniform membrane E_{kin} vs time is a peak identical to the original pulse, reaching a given x-position at time $t=x/v$, with v the wavepacket's velocity. In the nanostructured membrane, the wavepacket is scattered in different directions by the holes' pattern, redistributing the vibrational energy. On one hand, this reduces the vibrational energy of the fraction of wavepacket that is not scattered and continues to propagate straight along x, leading to its attenuation, and thus a finite lifetime that we have fitted, τ_{coh} . On the other hand, the fractions of the original wavepacket scattered in other directions can recombine after having travelled longer (not straight) distances, giving raise to the presence of vibrational energy at later times. The fact that the kinetic energy at later times can show prominent maxima and minima is due to the interference among reflected wavepackets.

Finally, it is not clear if the back-scattering is a coherent specular reflection either (since the roughness of the side walls of the holes can't be measured). These reflections could just be incoherent, phase-destroying diffuse scattering off the pores, which cause significant resistance to heat flow and cannot realistically be modeled by a single adjustable τ_{BD} parameter, as far as I know. The observation that incoherent back scattering of phonons cause additional thermal resistance in nanoscale structures has been done previously by several research groups in the past.

As explained above, in the simulations there is no pores' walls roughness, therefore scattering (including backscattering) does correspond to coherent reflections. In the real sample there will be also incoherent diffuse scattering from roughness. We model the former using a single parameter τ_b , as it has already be done in the past (Maire Sc. Adv. 2017, Ref. 28 of our manuscript). This is clearly an approximation.

Still, in our opinion the fact that we can almost perfectly reproduce the experimental data by adding to the behaviour observed in the not-structured membrane the coherent mechanism identified by the calculations proves that our simple modelling is justified. Indeed, only minor modifications of the roughness and anharmonicity parameter are needed to slightly improve the agreement with the experimental data.

On such ground we do believe that, at these wavelengths and in such low roughness samples, the incoherent contribution to the phonon lifetime is less relevant than the one associated with

τ_{coh} , and for this reason we can satisfactorily model the former with a single adjustable parameter. We remark again that the values for the effective roughness obtained by the fit are in agreement with the expectations, thus endorsing our simplified choice for modelling incoherent surface scattering. We finally mention that, as shown in the figure below, at our smallest wavelength, 55 nm, even with a roughness of 2 nm (3 times our estimated value) we would have a majority of specular reflection and thus coherent character (>80%).

Figure 1 Specularity parameter p , representing the fraction of specularly reflected phonon wave as a function of phonon wavelength λ for given roughness values η , as indicated in the legend

Reviewer #3 (Remarks to the Author):

I thank the authors for taking the time to address my concerns in the second round of the review. As pointed out by the first reviewer, some of the explanations are not strongly backed by evidence, and the authors acknowledge that there are a few important considerations that could not be addressed with the available data or due to the limitations of the experimental technique.

However, I do appreciate the novelty of the experimental technique and the attempt to extract important information on the phonon properties in amorphous membranes. I must also thank the authors for including short discussions in the manuscript/supplemental material on the questions that cannot be addressed with the available data/due to the limitations of the experiment.

Therefore, I believe this work can be accepted for publication in Nature Communications.